# Multi-Connectivity for 5G Networks and Beyond: A Survey

**DOI:** 10.3390/s22197591

**Published:** 2022-10-07

**Authors:** Tidiane Sylla, Leo Mendiboure, Sassi Maaloul, Hasnaa Aniss, Mohamed Aymen Chalouf, Stéphane Delbruel

**Affiliations:** 1COSYS-ERENA Lab., Université Gustave Eiffel, 33400 Talence, France; 2GEII-ISA, University of Sciences, Techniques and Technologies of Bamako, Bamako, Mali; 3Computer Sciences & Mathematics, JUNIA Engineering School, 59000 Lille, France; 4IRISA Lab., University Rennes 1, 22300 Lannion, France; 5LaBRI, University of Bordeaux, 33400 Talence, France

**Keywords:** 5G networks, multi-connectivity, user association, quality of service, energy efficiency, fairness, mobility management, spectrum management

## Abstract

To manage a growing number of users and an ever-increasing demand for bandwidth, future 5th Generation (5G) cellular networks will combine different radio access technologies (cellular, satellite, and WiFi, among others) and different types of equipment (pico-cells, femto-cells, small-cells, macro-cells, etc.). Multi-connectivity is an emerging paradigm aiming to leverage this heterogeneous architecture. To achieve this, multi-connectivity proposes to enable UE to simultaneously use component carriers from different and heterogeneous network nodes: base stations, WiFi access points, etc. This could offer many benefits in terms of quality of service, energy efficiency, fairness, mobility, and spectrum and interference management. Therefore, this survey aims to present an overview of multi-connectivity in 5G networks and beyond. To do so, a comprehensive review of existing standards and enabling technologies is proposed. Then, a taxonomy is defined to classify the different elements characterizing multi-connectivity in 5G and future networks. Thereafter, existing research works using multi-connectivity to improve the quality of service, energy efficiency, fairness, mobility management, and spectrum and interference management are analyzed and compared. In addition, lessons common to these different contexts are presented. Finally, open challenges for multi-connectivity in 5G networks and beyond are discussed.

## 1. Introduction

One of the main objectives of fifth-generation cellular networks (5G) is to offer a solution to manage a growing number of users and an ever-increasing demand for bandwidth [1]. To achieve this, two main approaches have been considered so far. The first approach is to deploy low-power 5G base stations (pico-cells, femto-cells, small-cells) that overlay the macro-cellular network and share its licensed spectrum [2]. In this layered architecture, traffic can be offloaded from macro-cells to femto-/small-cells. The second approach is to offload traffic from the cellular network to other networks [3]: Wireless Fidelity (WiFi) networks, Low Earth Orbit (LEO) satellites, Low Power Wide Area (LPWA) networks, etc. These heterogeneous 5G networks (HetNets), whether they combine different radio access technologies (cellular, satellite, Wifi, etc.) or different types of equipment (pico-cells, femto-cells, small-cells, macro-cells, etc.), have the potential to significantly increase network capacity.

The idea of 5G HetNets and the resulting densification of the network has also led to the emergence of a new concept: Dual Connectivity (DC) [4] or multi-connectivity (MC) [5]. This approach was first introduced in the Third-Generation Partnership Project (3GPP) Release 11 [6] and Third-Generation Partnership Project (3GPP) Release 12 [7]. It aims to enable User Equipment (UE) to be simultaneously connected to a macro-cell base station, usually called a master node, and to one—dual connectivity—or more—multi-connectivity—lower-tier node(s) (pico-cells, femto-cells, small-cells) and/or heterogeneous node(s) (satellite, WiFi access point, etc.), usually called a secondary node(s) (cf. Figure 1).

Similar to Carrier Aggregation (CA) [8], the DC and MC approaches enable UE to simultaneously use radio resources on multiple component carriers. Thus, the amount of bandwidth experienced by the UE is increased. However, the benefits of multi-connectivity are not limited to this. Indeed, unlike CA, with MC/DC, a UE is simultaneously connected to different network nodes and not only one. Using component carriers from different network nodes, it could be possible to ensure the efficient management of the mobility of the UE without any disconnections [9]. It could also be possible to enhance reliability, in particular for Ultra-Reliable Low-Latency Communication (URLLC), as the same information could be sent through different channels [10]. Finally, several research projects are investigating the potential of MC to reduce the power consumption of the 5G network [11,12] and to improve spectrum and interference management [13].

Multi-connectivity is a new paradigm and different survey papers dealing with user association in 5G networks do not consider this approach [14,15,16,17]. As a result, few survey papers have so far tackled this issue [4,10,18,19]. Most of these papers [4,10,18] present two main limitations. First, they only focus on the combination of different equipment and do not include the combination of different Radio Access Technologies (RATs). Thus, they only provide a partial overview of multi-connectivity in 5G networks and beyond. In addition, they do not introduce any existing work and only focus on the presentation of the functioning of the DC/MC approaches. The study proposed in [19] presents a broader vision of existing work but focuses only on the improvement of communication reliability.

This is why, compared with existing survey papers, our work presents several new aspects and provides a basis for future work on multi-connectivity in 5G and future networks. The main contributions of this paper are:A comprehensive review of existing standards and enabling technologies, highlighting their characteristics and benefits;The definition of a taxonomy enabling the classification of the different elements characterizing multi-connectivity (strategy, objective, RATs, etc.) in 5G HetNets;A comprehensive comparison of existing MC-based applications in 5G networks and beyond aiming to improve Quality of Service (QoS), energy efficiency, fairness, mobility, and spectrum and interference management;A summary of the lessons that can be learned from existing research;An identification of open challenges and future directions in the field of multi-connectivity for 5G and future networks.

The rest of this paper is organized as follows: Section 2 offers a brief introduction of existing standards and enabling technologies for multi-connectivity. Then, in Section 3, the proposed taxonomy is introduced and described. In Section 4, the state-of-the-art applications (QoS, energy efficiency, fairness, mobility management, spectrum and interference management) of multi-connectivity in 5G networks and beyond are presented and analyzed. In addition, possible improvements to these applications are identified. Lastly, the most relevant remaining challenges for multi-connectivity in 5G and future networks are discussed in Section 5.

## 2. Existing Solutions for Multi-Connectivity Deployment

This section aims to present existing standards and enabling technologies for MC in 5G networks and beyond.

### 2.1. Existing Standards

In this section, existing standards for multi-connectivity in 5G and future networks are presented (see Table 1). As a reminder, in a multi-connectivity scenario (cf. Figure 1), a UE is simultaneously connected to at least two network nodes, a macro-cell base station, and a lower-tier node (pico-cells, femto-cells, micro-cells, small-cells) or a heterogeneous node (satellite, WiFi access point, LPWA gateway). With multi-connectivity, this UE is able to simultaneously use radio resources on both component carriers, aggregating bandwidth or/and improving reliability [5].

Existing standards focus on three main types of multi-connectivity scenarios, a Long-Term Evolution (LTE)-LTE DC/MC or LTE-5G New Radio (NR) DC/MC (cf. Section 2.1.1 and an LTE-Wireless Local Area Network (WLAN) DC/MC (cf. Section 2.1.2).

It should be noted that the standards described in this section are in the development phase. Thus, many improvements are still required, such as latency reduction, congestion control, and reliability [20]. Among the solutions considered to meet these challenges, both for LTE-LTE/LTE-5G NR multi-connectivity and LTE-WLAN multi-connectivity, the MultiPath-Transmission Control Protocol (MP-TCP) approach is widely studied [21,22,23]. Indeed, this protocol, enabling a TCP connection to use multiple paths simultaneously, could improve the quality of the experience for the user and the performance of the network in a DC/MC scenario [23].

It should also be noted that although standards enabling multi-connectivity with satellite and LPWA RATs have not yet been defined, the integration of these technologies within 5G cellular networks and beyond is currently being considered in both industrial and research environments [24].

#### 2.1.1. LTE-LTE/LTE-5G NR Multi-Connectivity

So far, two main standards have been designed to enable LTE-LTE multi-connectivity: the LTE-Coordinated Multi-Point (CoMP) standard and the LTE-DC standard. These technologies supporting 5G NR RAT and LTE-5G NR multi-connectivity were then extended, respectively, to the 5G-CoMP standard and the E-UTRAN-NR Dual Connectivity (EN-DC) standard.

##### CoMP Standard

The CoMP standard, first introduced in 3GPP Release 11 [6] and then extended in 3GPP Release 12, was the first approach standardized by the 3GPP to enable multi-connectivity in cellular networks. This approach assumes that a UE in a cell-edge region can transmit/receive data to/from several base stations. LTE-CoMP was designed to enable the dynamic coordination of the transmission and reception of data between a UE and the LTE base stations. Indeed, this could improve both network coverage and network utilization while reducing inter-cell interference [25]. With this protocol, the same component carrier for both cells, as well as a single Radio Link Control (RLC) entity and a single Medium Access Control (MAC) entity, are used. The CoMP technology is based on the coordination of the different nodes through direct exchanges using the X2 backhaul interface: coordination of the scheduling (UpLink (UL)/Downlink (DL)), coordination of the beam-forming (DL), joint reception (UL), joint processing (UL/DL) and Inter-Cell Interference Coordination (ICIC) [25]. This coordination is enabled by the exchange of different pieces of information between the base stations: load level, reference signal receive power measurements, etc. A metric evaluating the trade-off between the costs and the benefits of multi-connectivity has also been used to define an optimal solution [26]. With the introduction of the 5G NR RAT in 3GPP Release 16, the CoMP protocol has been extended to support LTE-5G NR multi-connectivity (5G-CoMP). For multi-connectivity between different RATs, the 5G-CoMP protocol is based on the same principle but includes improvements that should ensure higher data rates and lower latency [27,28]: CoMP spatial multiplexing, CoMP spatial diversity, etc.

##### DC Standard

The DC standard, first introduced in 3GPP Release 12 [29] and then extended in all 3GPP releases, was a second approach standardized by the 3GPP to enable multi-connectivity in cellular networks [30]. DC and CoMP standards have the same objective (i.e., to connect a UE to several base stations) and are based on direct exchanges between these base stations through the X2 interface. Nevertheless, there are many differences between these two technologies. First, the DC standard implies the definition of different roles for the base stations involved in the multi-connectivity process. Indeed, one of these base stations is considered the master node, managing the control plane connection, whereas the other is considered the secondary node. Then, unlike the CoMP protocol, with DC, the UE uses a specific component carrier for each base station rather than the same one, enabling carrier aggregation. Finally, this UE manages a specific RLC and MAC entity for each connection with a base station. This means that radio resources are provided by different radio schedulers (one for each base station) and that this UE must support multiple UL carriers. This approach, considering non-ideal backhaul X2 interfaces, aims to provide higher data rates as well as lower latency than the CoMP approach [31]. Nevertheless, it requires more efficient interference coordination. With the introduction of the 5G NR RAT in 3GPP Release 16, the LTE-DC standard was extended to support LTE-5G NR multi-connectivity (EN-DC). The main objective of the EN-DC protocol is to manage the different 5G NR deployment phases [32]. For each of these phases, Non-StandAlone (NSA) using the 4G core network and StandAlone (SA) using the 5G core network, specific control solutions were proposed. It can be added that as the user plane can be split in both the core network and macro-cell base station for both the LTE-DC and EN-DC standards, different architectures/configurations can be considered for the user and control planes [4].

#### 2.1.2. LTE-WLAN Multi-Connectivity

So far, two main standards have been designed to enable LTE-WLAN multi-connectivity: the LTE-WLAN Aggregation (LTE-LWA) standard and the LWA with IPsec Tunnel (LTE-LWIP) standard. In future 3GPP releases, these technologies should be extended to support 5G NR-WLAN multi-connectivity. It should be noted that these technologies are usually associated with the LTE-License-Assisted Access (LTE-LAA) [33], LTE-Unlicenced (LTE-U) [34], and MulteFire [35] standards. Indeed, all these technologies focus on the interaction between cellular and WiFi networks. However, they consider different solutions. Unlike the LTE-LWA and LTE-LWIP approaches, the LTE-LAA, LTE-U, and MulteFire solutions aim to enable cellular operations in unlicensed bands. Therefore, they are not addressing LTE-WLAN multi-connectivity but rather the fair coexistence of cellular and WiFi networks in unlicensed bands, mainly through the definition of listen-before-talk access schemes [36]. For this reason, these standards are not presented in this section.

##### LTE-LWA

The LTE-LWA standard, first introduced in 3GPP Release 13 [37] and then extended in Release 14 (Enhanced LWA—eLWA) to support WiGig, was the first approach standardized by the 3GPP to enable LTE-WLAN multi-connectivity. The LTE-LWA solution operates similar to the LTE-DC standard. Indeed, in a LTE-WLAN scenario, with LTE-LWA, data can be transmitted via both the Wifi bearer and the LTE bearer and the control plane connection is managed by the LTE base station, corresponding to the master node. To enable this, a new element was added to the network architecture by the 3GPP, the Wireless Termination (WT). This logical element can be integrated within WiFi Access Points (APs) or into WiFi Access Controllers (ACs). It establishes a connection between the base station and Wifi APs to manage the data aggregation process and exchanges (control/data) between LTE and Wifi devices. With this architecture, LTE-LWA provides per-packet routing between the base station and the AP at the Packet Data Convergence Protocol (PDCP) level. This guarantees the efficient control and utilization of resources on both links and improves system capacity and perceived throughput. It can be added that two main LTE-LWA deployment scenarios were considered: a collocated scenario (integrated base station and AP) and a non-collocated scenario (different physical locations), requiring the deployment of standardized Xw interfaces [38].

##### LTE-LWIP

The LTE-LWIP standard, first introduced in 3GPP Release 13 [39], was a second approach standardized by the 3GPP to enable LTE-WLAN multi-connectivity. The main objective of this solution is to offer LTE-WLAN multi-connectivity without requiring any modifications of the WLAN infrastructure. Unlike the LTE-LWA approach, with LTE-LWIP, a WT is not necessary and legacy WLAN devices can be deployed without any modifications. To do so, IPSec connections are established between the WiFi AP, the base station, and the UE. Then, the Radio Resource Control (RRC) link existing between the UE and the base station is used to perform per-packet bearer switching between the base station and the WiFi AP. Some part of the data is transmitted directly between the UE and the base station and another part is transmitted via the Wifi AP. Data are then aggregated above the PDCP level. However, this solution does not support split bearers. This results in lower performance than the LWA approach [37]. It can be added that, similar to the LTE-LWA approach, collocated and non-collocated scenarios were considered for this LTE-LWIP technology [37].

### 2.2. Enabling Technologies

In this section, different enabling technologies for multi-connectivity in 5G and future networks are presented: Software-Defined Networking (SDN), Network Function Virtualization (NFV), Cloud-Radio Access Network (C-RAN), Cognitive Radio (CR), Network Slicing (NS), and Artificial Intelligence (AI). Table 2 provides a synthetic overview of these technologies supported by 5G cellular networks or expected to be supported by future cellular generations [40].

It should be noted that, although these technologies are presented individually, they form a coherent system that can improve cellular networks. Indeed, SDN and NFV technologies, usually combined [41], can be used together to support the C-RAN approach [42]. In the same way, this system (SDN, NFV, C-RAN) combined with CR provides an architecture that enables the efficient deployment of NS [43]. Finally, AI techniques, among other things, can be applied to this NS framework [44].

#### 2.2.1. SDN

Software-Defined Networking [45] is the first enabling technology for MC in 5G networks and beyond. SDN is a network architecture that decouples the network control plane and the network forwarding plane [46]. In this architecture, data transfer is still performed by the network routers but the network control is performed by a logically centralized component: the SDN controller. This SDN controller has a centralized view of the network state and can program the routers’ behavior. Thus, it guarantees the global and efficient management of the network. In the MC context, the benefits of SDN could be numerous. First, SDN could be a way to limit the amount of control data required to manage multi-connectivity [47]. Then, as the controller has a global view of the network, SDN could provide the efficient management of user mobility and load balancing [48]. Finally, the SDN controller, using a standardized communication protocol, could enable interoperability between different RATs [49].

#### 2.2.2. NFV

Network Function Virtualization is the second enabling technology for MC in 5G networks and beyond. NFV is a network architecture concept aiming to decouple network functions (firewalls, traffic control, routing) from proprietary hardware appliances [50]. Thus, with NFV, network functions are turned into software functions running on virtual machines. This could lead to a significant cost reduction for multi-connectivity and more widely for cellular networks, as network functions could be deployed on any low-cost hardware (X86 server for example). Virtualization also guarantees a significant level of scalability. Indeed, resources (storage, computation, networks) could be allocated to a given network function according to its actual needs [51]. Finally, in an MC scenario, NFV could be used to move network functions according to the position of users and thus guarantee a high QoS [52].

#### 2.2.3. C-RAN

Cloud-Radio Access Network is the third enabling technology for MC in 5G networks and beyond. C-RAN is an evolved architecture for cellular networks that decouples Baseband Units (BBUs) from radio access units [53]. With C-RAN, these BBUs that manage network resource allocation are grouped in the cloud, forming a centralized BBU pool. This pool of BBUs, connected over optical fiber to Remote Radio Heads (RRHs), enables the real-time collaborative management of the network nodes. The benefits of this architecture for multi-connectivity in 5G and future networks are numerous. First, Heterogeneous C-RAN (H-CRAN) appears to be a promising solution to manage different RATs. Indeed, H-CRAN could enable the integration of heterogeneous RATs into a global cloud architecture [54]. Then, C-RAN could support a scalable deployment of RRHs [42]. Thus, with C-RAN, a high number of small-cell and femto-cell Base Stations (BSs) could be considered, increasing the number of multi-connectivity options. Finally, this architecture reduces both operational and capital expenditure [55].

#### 2.2.4. Cognitive Radio

Cognitive Radio is the fourth enabling technology for MC in 5G networks and beyond. CR corresponds to transmitter and/or receiver equipment that can sense available network channels and modify its parameters (modulation, transmission power, frequency bands) to use the channel that best meets its needs [56]. As this approach is user-centric, it was initially mainly studied in ad hoc networks. Indeed, this solution is in contrast to the current functioning (centralized) of infrastructure-centric cellular networks. Nevertheless, CR is now promoted by network operators. It could be an efficient way to improve cellular networks’ scalability and to efficiently share network resources among users [57]. Thus, CR could solve the spectrum scarcity problem related to the ever-increasing use of mobile services [58]. Moreover, in a multi-connectivity scenario, this approach could also be used to manage MC from a user-centric perspective as it could enable user equipment to select an optimal channel for a given application [59]. Therefore, CR could provide a higher level of flexibility.

#### 2.2.5. Network Slicing

Network Slicing is the fifth enabling technology for MC in 5G networks and beyond. NS is an evolved architecture for cellular networks that enables the deployment of end-to-end virtualized and independent logical networks over a single physical network infrastructure [43]. Thus, the objective of NS is to provide a QoS adapted to each service without requiring the deployment of multiple physical networks. NS, which abstracts hardware devices and virtualizes network functions and resources, can be seen as a combination of SDN and NFV (combined benefits of both these technologies) [60]. Therefore, this approach provides the benefits of both approaches: cost reduction, mobility management, load balancing, interoperability, scalability, and overhead reduction. Moreover, as NS creates end-to-end virtual networks, it would be possible to independently manage different applications. Therefore, in a multi-connectivity scenario, NS could be a way to improve flexibility [61]. Indeed, each application could be associated with the network node that best meets its needs in terms of QoS, security, etc. It can be noted that this approach has already been applied to access networks other than classic cellular networks, for example, in the context of WiFi networks [62,63].

#### 2.2.6. Artificial Intelligence

Artificial Intelligence is the sixth enabling technology for MC in 5G networks and beyond. AI corresponds to a set of decision support tools [64]: machine learning, deep learning, fuzzy logic, etc. The operation of these tools differs but their fundamental principle is the same. They enable a system to learn from experience (past behavior) and to perform human-like tasks (adaptation to current behavior). Thanks to the emergence of low-cost equipment with significant storage and calculation capacities, AI techniques can now be applied to large volumes of data [65]. This has enabled reliable predictive models to be designed and extended to complex decision-making processes. This is why AI appears today as a solution to many problems, in particular in cellular networks [64] for resource placement, communication path calculation, etc. AI could therefore be applied to SDN and NFV [66], as well as to C-RAN [67], CR [68], and NS [44]. In a multi-connectivity scenario, AI could ensure better mobility and spectrum management and a more efficient load balancing process [69] by adapting in real-time the network devices’ behavior to the network state and position of the UE.

## 3. Taxonomy

This section presents the taxonomy that we defined for the investigation of MC in 5G networks and beyond. This taxonomy provides a framework for the systematic review of state-of-the-art solutions. In the remainder of this paper, it is used to efficiently and objectively identify the advantages and disadvantages of the mechanisms/architectures/algorithms proposed so far in the literature. It is also intended to be reemployed by other researchers interested in analyzing existing work on multi-connectivity.

As shown in Figure 2, the proposed taxonomy is composed of seven main components: aim, considered RATs, type of heterogeneity, approach, topology, control, and selection process.

### 3.1. Aim

The first important element of this taxonomy is the goal associated with the use of MC in 5G and future networks. In the literature, we have identified five main targets: quality of service, energy efficiency, fairness, mobility management, and spectrum and interference management. It should be noted that in the literature, some papers focus on only one of these aspects, whereas others focus on several aspects (cf. Section 4).

#### 3.1.1. Quality of Service

Improving the quality of service of cellular networks is the first objective associated with the use of MC. QoS refers to the performance level offered by a network for a specific service. It corresponds to a certain service level perceived by the user. The definition of a minimum performance level guarantees the proper functioning of the service while optimizing the operator’s costs [70]. Different QoS parameters can be considered, as shown in [71]. So far, those that have been considered in MC literature are associated with performance such as end-to-end latency and throughput, jitter, reliability (packet loss), and availability.

#### 3.1.2. Energy Efficiency

Improving the energy efficiency of cellular networks is the second objective associated with the use of MC. Indeed, by reducing energy consumption, the environmental impact of cellular networks can be limited. This evolution toward green cellular networks is a major and crucial challenge today [72,73]. In addition, from an operator’s perspective, this could cut costs associated with network management. To achieve this goal, two main approaches have been considered in the MC literature: the effective management of energy-intensive handovers and optimal management of the user–cell association. The authors of [74], for example, have shown the value of MC in reducing energy consumption.

#### 3.1.3. Fairness

Guaranteeing fair access to network resources is the third objective associated with the use of MC. This, particularly in scenarios with limited radio resources, aims to ensure that each user has fair access to the network, regardless of the quality of their communication channel. The fair allocation of network resources is usually based on the estimated total available bandwidth and the number of users requiring access to the network [17].

#### 3.1.4. Mobility Management

Improving users’ mobility management is the fourth objective associated with the use of MC. Effective mobility management must allow continuous access to the network and services for the users wherever they are. Multi-connectivity could reduce the problems associated with mobility (latency, packet loss, etc.) as explained by the authors of [10].

#### 3.1.5. Spectrum and Interference Management

Improving spectrum and interference management is the fifth objective associated with the use of MC. Spectral efficiency aims to maximize the amount of data transmitted over a given frequency spectrum (capacity) as well as the coverage area. The reduction of interference is a possible solution to achieve this [75]. In the MC literature, different papers such as [13] have demonstrated how multi-connectivity could improve spectrum efficiency through the use of different base stations for UL and DL traffic management.

### 3.2. Considered Topology

The network topology considered for the definition of MC-based solutions is the second important element characterizing MC in 5G networks and beyond. Usually, two types of network models are considered [17]. First, uniform models, especially grid models, are widely used. In these models, network nodes are considered to be uniformly distributed on a regular grid. Random models (Poisson, Bernouilli) are also used frequently for network modeling. To demonstrate the relevance of one MC-based solution, several works have demonstrated that the defined solutions provide good performance for these two types of models (random and grid). However, for future 5G networks (integrating MC capabilities), it could be relevant to define new models specifically designed for this environment. Indeed, considering an existing deployment of macro-cells base stations, it could be necessary to determine an optimal deployment of the other network nodes to maximize QoS, energy efficiency, spectral efficiency, and user mobility management. In other words, it might be useful to assess the impact of multi-connectivity on the optimal deployment for 5G and future networks.

### 3.3. Considered Radio Access Technologies (RATs)

The third important element characterizing MC is the Radio Access Technologies that could be used to enable this multi-connectivity in 5G networks and beyond. From the literature, we have identified four main types of RATs: cellular (LTE, 5G NR), WiFi (IEEE 802.11ac, IEEE 802.11p), LPWA (NB-IoT, LoRA), and LEO satellite communications. As these technologies have different characteristics (deployment, coverage, performance, security), they could not be used for the same cellular services. Indeed, for applications requiring high bandwidth (Enhanced Mobile Broadband) or low latency and high reliability (Ultra Reliable Low-Latency Communications), cellular and WiFi technologies are the most suitable. For these applications, satellite communications could also be considered in certain scenarios [76] such as uncovered areas, limited capacity, etc. In contrast, for direct communications between devices (massive machine-type communications), LPWA and satellite technologies would seem appropriate [77]. Thus, the identification of RATs to be used in a given scenario is an important question (cf. Section 4.5).

### 3.4. Type of Heterogeneity

The type of heterogeneity considered is the fourth important element characterizing MC in 5G networks and beyond. MC corresponds to a situation where a UE is simultaneously connected to at least two network nodes and can simultaneously use radio resources on both component carriers [5] (cf. Figure 1). According to the 3GPP definition, one of these nodes, usually named the master node, is a macro-cell base station. In contrast, the secondary node can be either a lower-tier node (pico-cells, femto-cells, micro-cells, small-cells) or a heterogeneous node (satellite, WiFi access point, LPWA gateway, LTE/5G NR). Consequently, there are two types of heterogeneity, technological heterogeneity if the secondary node corresponds to a heterogeneous node using another RAT and equipment heterogeneity if the secondary node corresponds to a lower-tier node. It should be noted that when the master node is a 4G node and the secondary node is a 5G node, the heterogeneity could be twofold, that is, different equipment and different RATs.

### 3.5. Strategy

The strategy used for data transmission in a multi-connectivity scenario is the fifth element characterizing MC in 5G networks and beyond. Indeed, as the UE is simultaneously connected to different network nodes, for each application or even each packet, the UE must determine which node should be used to transmit data. Usually, the selection process is used to determine which network node would best meet the requirements of the application at any given time such as latency, bandwidth, packet loss, etc. However, in specific situations requiring a high level of reliability, for example, for URLLC applications [78], it may be necessary to transmit information simultaneously to both network nodes to guarantee that the information is effectively received. Thus, different strategies can be considered according to the scenario (mobility, availability, reliability, etc.) and the needs of the application(s) such as a single transmission, through a single node, or a multiple transmission through several nodes.

### 3.6. MC Control

The mechanism considered for MC control is the sixth element characterizing MC in 5G networks and beyond. Indeed, without an efficient control mechanism, the proper functioning of MC will not be guaranteed such as the user–cell association, mobility management, data transfer, etc. In the literature, there are three main approaches for communication control. First, there is the UE-centric approach [79]. Under this solution, the UE controls its association with the network infrastructure. Using the information gathered from the different RATS and a selection algorithm, the UE determines which communications should be established with the network nodes at any point in time. In contrast, in the infrastructure-centric approach [80], the network infrastructure controls the user–cell association. This can be done at the macro-cells’ base station level in a distributed approach, or at a higher level using a central SDN controller [81] in a centralized approach. However, as noted by the authors of [80], the implementation of infrastructure-centric solutions can be complex in 5G HetNets integrating heterogeneous RATs. This is the reason that today, hybrid approaches [82] are largely adopted. In these approaches, the decision is usually performed at the macro-cell base station level, using not only the information provided by the other base stations but also the information transmitted by the UE such as the available networks, quality of service, etc. Thus, in this approach, the perception of the 5G network is improved thanks to exchanges between the UE and infrastructure.

### 3.7. Tools

The tools used to manage MC control-related issues are the seventh element characterizing MC in 5G networks and beyond. These tools are useful at several levels that require an automatic decision-making process such as the user–cell association [83], the RAT selection [84], and the data transmission strategy (single/multiple). Existing tools are usually divided into two main categories: hard-computing and soft-computing techniques [85]. Hard-computing techniques correspond to traditional solutions such as symbolic logic and utility functions. These approaches, based on binary logic, require an exactly solvable state analytic model. In a real-world and evolving environment, defining such models is complex or even impossible. Therefore, the application of these tools in highly variable environments is not necessarily relevant so in 5G networks, existing research works such as [83,86] are usually based on soft-computing techniques such as fuzzy logic, evolutionary computing, probabilistic reasoning, machine learning, neural networks, support vector machines, game theory, etc. Indeed, as these techniques are error-tolerant, they provide a higher level of robustness as well as an ability to cope with unexpected scenarios [87]. These soft-computing techniques are built upon two main principles: adaptivity (adaptation to new contexts) and knowledge (ability to learn). It should be noted that soft-computing techniques can be applied to hard-computing techniques to couple the advantages of these two approaches [88], that is, accuracy, quick decision making, low computational overheads, and low costs.

## 4. Existing Applications of Multi-Connectivity in 5G Networks and Beyond

With the development of several standardized solutions (cf. Section 2.1) and the emergence of 5G cellular networks promising the deployment of a large number of small-cell base stations, the interest in multi-connectivity has increased.

Five main objectives have been identified in papers discussing the use of multi-connectivity in cellular networks (cf. Section 3.1), that is, QoS improvement, energy efficiency, fair use of network resources, mobility management, and spectrum and interference management.

In this section, these research papers, classified by purpose, are introduced and compared. Moreover, for each identified target, the remaining challenges are highlighted.

### 4.1. Multi-Connectivity to Improve Quality of Service

The improvement of QoS is the first objective associated with the use of multi-connectivity in 5G and future networks (cf. Section 3.1.1). In this section, the existing solutions are compared (cf. Section 4.1.1) and analyzed (Section 4.1.2).

#### 4.1.1. Existing Solutions to Improve QoS

Various papers have already explored the use of MC to improve the QoS. More specifically, these papers have focused on four main topics: perceived throughput improvement [89,90,91], latency reduction [92,93,94], reliability [78,84,95,96], and availability improvement [97,98].

The improvement of the user’s perceived throughput is usually presented as one of the intrinsic benefits of multi-connectivity (bandwidth additions). Therefore, various papers have focused on this subject such as [89,90,91]. In this work, the authors of [89] introduced a new mechanism that aimed to maximize the overall system throughput for DL in 5G networks and beyond. To achieve this, they first formulated the optimization problem in a sum-of-ratios form. Then, to solve this non-deterministic polynomial hard (NP-hard) problem, they introduced an iterative algorithm based on the Karush–Kuhn–Tucker (KKT) conditions, which allowed them to define a sub-optimal solution. The evaluation carried out by the authors demonstrated the benefits of this solution in terms of throughput maximization. However, the scenario considered by the authors seems simplistic (single macro-cell BS, lack of mobility). Moreover, the inter-macro-cell BSs’ handover, which affects the performance, is not addressed in this proposal. Similarly, the authors of [90] tried to improve the throughput perceived by users. The main contribution of this paper was the definition of an architecture enabling the use of millimeter waves (high-throughput) in a multi-connectivity scenario while guaranteeing a high level of connectivity. To achieve this, the authors focused on the definition of an MC framework based on the existing standard, which should allow a UE to connect simultaneously to a 4G BS and a 5G BS supporting millimeter waves. The authors also introduced an algorithm for the optimal selection of secondary BSs that aimed to maximize connectivity for UEs. An important limitation of this work, which focused on the definition of an architecture integrating millimeter waves, was that it only considered non-standalone 5G architecture (and not standalone architecture). Therefore, it is impossible to measure the potential long-term benefits of the proposed solution. Moreover, the proposed selection algorithm was simplistic and inflexible (definition of fixed thresholds). The authors of [91] also looked at the use of millimeter waves in an MC scenario. In particular, to deal with the limitations of millimeter waves (range, frequent disconnections), they proposed a mechanism to solve the related link-scheduling problem. The proposed iterative algorithm was based on temporal decomposition. It aimed to maximize the throughput perceived by the users for each of these time windows while taking into account the power limits of the milliliter BS. The solution and experiments introduced in this paper were more comprehensive than those described in [90]. However, the approach described here appeared to be costly both in terms of control (exchange between devices) and management (number of handovers). It could therefore be interesting to evaluate the overheads associated with this approach and try to minimize them.

Latency reduction is the second potential benefit of the use of MC in 5G and future networks [92,93,94]. To achieve latency reduction, the authors of [92] have introduced an MC architecture enabling different RATS to be managed simultaneously: 4G LTE, 5G NR, Wi-Fi and WiGig. The overall management of this architecture is based on the use of an SDN controller. This should allow the selection of the most suitable RAT according to the user’s needs and the performance guaranteed by each RAT such as latency, bandwidth, etc. The main interest of this paper is that it proposes a software architecture that could allow fine-grained QoS management for each user or even each application. However, the proposed solution is still in the draft stage. Therefore, it could be interesting to use this proposal as a basis for defining a powerful software solution for MC scenarios. The authors of [93] sought to find a trade-off between latency reduction and resource utilization in a multi-connectivity scenario. To achieve that, they proposed to activate multi-connectivity only for users requiring it (critical applications). Thus, they defined a heuristic latency-aware MC configuration algorithm integrating a critical latency threshold enabling the determination of the packets for which the activation of multi-connectivity is relevant. This solution, limiting the use of multi-connectivity, naturally demonstrated significant gains in terms of the use of resources. However, this solution did not take into account the network load in the latency threshold calculation. Similarly, only one RAT was considered. Thus, it might be interesting to consider new parameters and a more complete architecture to improve the performance of the proposed solution. The authors of [94] provided a comprehensive evaluation of the performance of a 4G/5G cellular architecture including MC services applied to latency reduction. To achieve this, they used the Measuring Mobile Broadband Networks in Europe (MONROE) testbed and implemented the LTE-SQ (basic MC management) algorithm with different configurations such as preferred RAT, number of simultaneous connections, etc. This paper focused on the performance evaluation of MC and described a framework and a set of measurements that could be used as a starting point for further work in this field. Thus, rather than limitations, this paper provided areas for improvement such as user preferences, network history, etc.

Reliability improvement also seems to be a potential benefit of multi-connectivity [78,84,95,96]. Reliability is generally improved by duplicating the packets transmitted by/to the UEs [78,95,96]. Thus, for critical applications, each packet is simultaneously transmitted to different BSs (primary/secondary) thanks to multi-connectivity. These different papers have the same objective, that is, to upgrade the standardized MC architecture to enable an efficient duplication of packets at the master base station level and, more specifically, at the PDCP (Packet Data Convergence Protocol) level. In addition, these papers, which focus on a theoretical definition of architecture, raise various questions related to the optimization of the RLC layer for PDCP duplication and the handover management in a duplication scenario. Going further, the authors of [84] proposed a machine learning-based RAT selection that aimed to improve reliability in an MC scenario. In this solution, packet duplication was carried out by taking into account the performance of each of the RATs and the history of the users. Thus, packets were duplicated only when necessary (reliability of the different RATs at a given time for a given user) and by selecting the RATs that guaranteed the highest level of reliability. This work was the first contribution that aimed to improve the reliability of communications through MC. However, the authors considered in this paper that all users have the same QoS requirements (homogeneous traffic). Moreover, the overheads associated with this machine learning-based solution (computation, latency) have not been evaluated. Therefore, many challenges remain in this field.

Finally, improved availability is the last potential benefit of multi-connectivity. However, similar to reliability improvement, this topic has not been studied extensively so far and only two papers provide relevant solutions [97,98]. The solution proposed in [97] focused on the availability of services (slices) at the network edge. Specifically, the authors aimed to ensure that a given service (slice) needed by a given user was available at the primary and secondary BSs to which this user could connect. To achieve this, they proposed a handover algorithm that aimed to ensure the transfer of the user’s data and required services. This mechanism, intended to be integrated into the 3GPP reference architecture, is simple. However, this paper introduced an interesting idea, that is, the dynamic management of slices in a multi-connectivity scenario. To support all the applications, it would seem necessary to consider different improvements, in particular, the pre-deployment of services according to the mobility of the users and the management of the users’ requirements (latency, bandwidth). The authors of [98] described availability as the ability of the network to meet the users’ needs in terms of latency, packet loss rate, and bandwidth. Therefore, they proposed a framework based on an exhaustive search method to maximize the communication link utilization and to select the most appropriate communication channel for each user/application. This more classical solution also offers interesting perspectives reliability improvement through MC. However, the scenarios considered for the evaluation seem to be unrealistic (architecture, mobility) and the framework defined in this paper is based on simple and inflexible mechanisms.

#### 4.1.2. Discussion Regarding QoS

Different papers have already tried to improve QoS in 5G networks and beyond through MC (see Table 3). As a result, the ideas of perceived throughput improvement [89,90,91], latency reduction [92,93,94], reliability [78,84,95,96], and availability improvement [97,98] have also been considered. Therefore, these papers, considering the different approaches, different RATS (cellular/Wi-Fi), and different tools (machine learning, heuristic, etc.), seem to be an interesting starting point for further research in this area. However, several limitations can be identified:First, it might be interesting to develop solutions that would jointly address the different QoS parameters (throughput, latency, jitter reliability, availability). Indeed, these parameters are intrinsically linked to each other. For example, solutions developed to improve reliability (packet duplication) will have an impact on network latency and available throughput. Similarly, maximizing the use of network capacity could have an impact on other QoS parameters. Thus, the simultaneous consideration of different QoS parameters would enable the definition of more efficient and effectively deployable solutions.A more specific but equally important point would be to develop more solutions to improve reliability and availability. Indeed, as mentioned in the previous section, work in these areas is still in its early stages. However, these parameters will be essential for future cellular applications and, in particular, URLLC applications. Moreover, the improvement of these parameters through MC opens the way to interesting solutions (optimal RAT selection, efficient packet duplication, mobility management, etc.).Finally, the issues related to the optimal management of services at the edge of the network (edge computing, slicing) have so far only been addressed in the context of availability improvement. However, due to the back-haul limitations, this optimal positioning of services seems essential in terms of latency, throughput, and reliability. Consequently, the management of services (mobility, positioning) should also be studied for these other parameters.

### 4.2. Multi-Connectivity to Improve Energy Efficiency

The improvement of energy efficiency is the second objective associated with the use of multi-connectivity in 5G networks and beyond (cf. Section 3.1.2). In this section, the existing solutions are compared (cf. Section 4.2.1) and analyzed (Section 4.2.2).

#### 4.2.1. Existing Solutions to Improve Energy Efficiency

Different research projects have already combined the ideas of energy efficiency improvement and multi-connectivity [11,12,99,100,101,102,103,104].

First, some researchers, considering that multi-connectivity could have a negative impact on the energy consumption of cellular networks, have proposed solutions that aimed to improve the energy efficiency of this technology [11,99]. Therefore, the authors of [99] have proposed a scheme to optimize both the user-cell association and the energy consumption of the network in an MC scenario. In particular, the authors defined a two-time-scale Lyapunov optimization mechanism enabling the efficient activation and deactivation of small-cell base stations depending on the queue and channel-state information of the UEs. Therefore, this solution aims to optimize both cell activation and user association to reduce energy consumption. The evaluation carried out demonstrates the gains of the solution in terms of energy consumption as well as its viability (low calculation times). However, two major limitations could be identified: the absence of a large-scale evaluation for the computational delays induced by the solution and the fact that the mobility of UEs was not considered (both in the definition of the solution and in the evaluation). Similarly, the authors of [11] focused on energy optimization in an MC scenario. However, the approach considered was completely different. Indeed, the proposed solution was based on three emerging technologies in the energy area: Energy Harvesting (EH), Energy Sharing (ES), and Wireless Power Transfer (WPT). The authors considered base stations powered by solar panels and introduced a Smart Energy Management (SEM) module that aimed to associate energy management and load balancing. Thus, they defined an optimization problem by taking into account the UE requirements and the amount of energy available for the small cell BS. This solution, combining two areas (network and energy), offered interesting perspectives. However, the solution defined was simple and could be optimized (delays, performance). Moreover, the architecture considered, powered mainly by solar panels, seems unrealistic and should have supported alternative options to ensure the proper functioning of the network.

On the contrary, other research studies have argued that multi-connectivity could improve energy efficiency in 5G and future networks [12,100]. Therefore, the authors of [100] sought to demonstrate how MC could improve the overall energy efficiency of 5G networks by decoupling UL and DL UE association. To do so, they first formulated the MC-based user–cell association optimization problem and proved that it was NP-hard. Then, they introduced an algorithm that aimed to solve the optimization problem in polynomial time by combining the LP Relaxation-Rounding (LPR-R) and Generalized Assignment Problem (GAP) heuristics. The evaluation provided by the authors confirmed the benefits of this solution in terms of performance (throughput) and energy consumption. However, this solution did not take into account the number of handovers between base stations, which is generally considered an important factor in energy consumption. Furthermore, beyond the split between UL and DL flows, it would have been interesting to look at the split between different applications (Network Slicing) by taking into account their respective requirements. The authors of [12] also attempted to demonstrate that MC could improve the energy efficiency of a network. To do this, the authors first defined different scenarios, a first scenario where MC did not offer any gains in terms of bandwidth (lower bound), a second scenario where the gain was weak (low performance bound), and a third scenario where the gain was important (upper bound). Then, they introduced five common algorithms to ensure an efficient user–cell association, that is, the max bitrate, max signal-to-interference-plus-noise ratio, max bitrate-energy efficiency, max clustered bitrate, and analytic hierarchy process. The experiments carried out in this paper demonstrated that these different MC algorithms could guarantee reduced energy consumption compared to a simple connectivity scenario. However, as this paper focused on the energy consumption assessment, it would have been interesting to take into account solutions optimizing energy consumption in a simple connectivity case and also to measure the impact of multi-connectivity at the UE level.

Finally, some research projects have focused on the definition of an optimal multi-connectivity solution that solves the trade-off between performance and energy consumption [101,102,103,104]. Therefore, the objective of the solution proposed by the authors of [104] was twofold and focused on guaranteeing acceptable delays while minimizing energy consumption by considering two main assumptions: (1) delays are induced by back haul and (2) small-cell BSs consume less energy than macro-cell BSs. Thus, they formulated a multi-objective optimization problem enabling them to meet both objectives through an estimation of the capacity of the base stations (small-/macro-cell) and the requirements (latency/bandwidth) of the users. The authors then tried to demonstrate the benefits of their solution but the scenario considered was too simplistic. Indeed, only one macro-cell BS and a limited number of interference-free small-cell BSs were deployed. Thus, inter-macro cell mobility and a more extensive study of the solution are necessary. The objective of the authors of [101,102] was very similar. Indeed, they also tried to optimize the global consumption of the network while taking into account the delays involved in inter-base station communications. However, the solution proposed here ([101,102]) was more advanced than the solution described in [104]. Indeed, the authors took into account the mobility of users between small-cell BSs and proposed an accurate estimation of the BSs’ load level based on an efficient exchange of information between UEs and BSs. Thus, this solution could offer an effective trade-off between performance and energy efficiency. However, it could be interesting to assess the impact (latency, energy consumption) of the communications defined for the estimation of the base station load, both at the UE and base station level. The solution proposed by the authors of [103] also focused on the definition of a trade-off between performance and energy efficiency. Similar to the authors of [11], the authors of [103] considered the use of solar panels to power small-cell BSs. However, in this paper, an important element that was not addressed in [11] was taken into account, that is, the possible performance degradation associated with the use of solar-panel-powered BS. Therefore, in this article, the user–cell association was not only dealing with energy consumption but also with the performance required by users. To enable this, a layered algorithm using the Karush–Kuhn–Tucker (KKT) conditions and providing a joint optimization of the traffic scheduling and power allocation was introduced. The proposed solution seems to offer interesting performances but could go further by considering a dynamic management of the energy produced by the solar panels (variation according to time and demand) and by taking into account the idea of energy sharing introduced in [11].

#### 4.2.2. Discussion Regarding Energy Efficiency

Different papers have already addressed jointly the ideas of energy consumption and multi-connectivity. As a result, some papers have sought to reduce the impact of multi-connectivity [11,99], whereas others have demonstrated its benefits [12,100] or attempted to trade off performance and energy efficiency [101,102,103,104]. These papers, considering different approaches and different tools (cf. Table 4), seem to be an interesting starting point for further research in this area. However, several limitations can be identified:First, it might be very interesting to look at the use of multiple RATs in this context. Indeed, existing papers have focused on the use of cellular technologies, whereas other technologies (WiFi, Bluetooth, etc.) could potentially reduce the overall energy consumption of the network thanks to multi-connectivity [105]. Performance evaluation and definition of new mechanisms for these heterogeneous architectures with the objective of energy consumption reduction would appear to be a relevant topic of study.In the same way, it would be interesting to determine the optimal positioning of the master and secondary nodes for MC scenarios, in the case of both pure cellular and heterogeneous networks. Indeed, as noted by the authors of [106], an efficient architecture could lead to a significant reduction in network consumption and there are many possibilities for optimization in this area as multi-connectivity is an emerging concept.Finally, in line with the proposition introduced in [100], it could be interesting to look at the use of software/centralized approaches in this context. Indeed, C-RAN architectures (II.B.3) represent the future of cellular network management and could, perhaps, offer better management of mobility and load balancing. Consequently, it might be interesting to apply this idea to energy management.

### 4.3. Multi-Connectivity to Improve Fairness

Fair use of the network is the third objective associated with the use of multi-connectivity in 5G and future networks (cf. Section 3.1.3). In this section, the existing solutions are compared (cf. Section 4.3.1) and analyzed (Section 4.3.2).

#### 4.3.1. Existing Solutions to Improve Fairness

Different research projects have combined the ideas of fairness improvement and multi-connectivity: [107,108,109,110].

For all these papers, the underlying idea is the same: improving fairness in user access while maximizing the total throughput utility. To achieve this, the proposed solutions are based on a widely used scheduling algorithm, the Proportional Fair (PF) algorithm, which aims to maximize the per-user throughput’s logarithmic sum (PF metric, [111]). However, this scheduling algorithm was designed for simple associations between UEs and base stations and not for a multi-connectivity environment. Moreover, UE association in the MC context has been proven to be NP-hard [107]. Therefore, each paper in the literature proposed an evolution of the PF algorithm that aimed to guarantee high performance and fairness in this context.

The solution introduced in [107] integrated two main improvements compared to the standard PF algorithm. First, the authors considered that the PF metric was, in an MC case, equal to the average throughput received by the UE from all the network nodes it was simultaneously connected to. They also considered that a centralized controller calculates the average throughput of all users, whereas a distributed approach is generally used by the PF algorithm. Then they defined three heuristic algorithms that should enable the association of UEs in a multi-connectivity context. These algorithms that require controller intervention, aimed to offload traffic from macro-cell base stations to small-/femto-cell base stations and to define for each UE, thanks to a stable matching scheme, two associations with base stations offering a maximum Reference Signal Received Power (RSRP). This system presented two main limitations. First, it was based on a non-standardized architecture that could be complex to deploy. Second, the additional costs (latency) potentially associated with this architecture were not evaluated.

The authors of [108], focused on LTE-WLAN aggregation and developed a similar solution. The main interest of this paper was the fact that different deployment scenarios were considered (ideal and non-ideal back haul). This ensured a high level of performance in realistic scenarios. The defined algorithm was simple and reproduced a “Water-Filling” mechanism. The amount of data transmitted by the UE to the macro-cell base station was proportional to the UE peak capacity ratio for this macro-cell base station and the throughput of this UE for the small-cell base station/WiFi AP. The main limitation of this approach was that it was only reactive and therefore did not predict UE behavior, inducing latency. Moreover, the deployment of this solution within the reference architecture was not studied and this integration could be complex.

The solution proposed in [109], introduced a new constraint compared to the scenarios considered in [107,108]. Indeed, the idea of minimum rate requirements for each UE was proposed, enabling network resources (bandwidth) reservation. In addition to this parameter, the authors of this paper also introduced a new optimization method to improve fairness. This two-level iterative algorithm was based on a Lagrange dual decomposition method integrating specific Lagrange multipliers enabling the management of traffic split and rate constraints. This alternative guaranteed a low level of complexity but considered a scalar channel model, ignoring small-scale fading effects and resulting in inaccurate calculations. Moreover, the considered scenario was simplistic and far from a real-work environment in terms of user density, base station positioning, etc.

The approach defined in [110] was another solution proposed in the literature that aimed to improve fairness using multi-connectivity. To solve the non-convex cell-association problem, the authors, considering a distributed architecture, introduced a matching game algorithm with externalities, enabling it to reach a local optimum. Moreover, to mitigate interference and improve fairness among users, they also defined a joint cell-selection and power-control algorithm. This iterative algorithm deals with power control and cell selection separately and sequentially. This solution presented interesting performances. However, it was weakly positioned compared to existing works and the reference architecture. Therefore, it could be complex to deploy. In addition, the authors only considered multi-connectivity between 4G base stations, limiting the scope of application of this method.

#### 4.3.2. Discussion Regarding Fairness

Research papers dealing with the use of multi-connectivity for improving fairness have introduced different implementable methods [107,108,109,110]. These papers, considering different tools (game theory, water-filling, decomposition), as well as different control solutions (centralized, distributed), provide an interesting starting point for future research in this field (cf. Table 5). However, several limitations can be identified:Even if different control solutions have already been considered, none of the existing works have proposed a comparison between a centralized and a distributed approach. This could help to determine an optimal solution in the MC context. Moreover, for centralized control, it could be interesting to consider the development of solutions based on SDN technology. Indeed, this standardized technology could be an efficient way to globally manage the network and share resources fairly among users.The idea of fair resource allocation among users has not been considered yet for 5G sliced networks. However, this idea could be interesting and could lead to a more complex model and additional constraints related to the QoS requirements of each of the user slices. Thus, in this Network Slicing context, enriched proposals could be defined.Although different tools have been considered so far, Artificial Intelligence techniques have not been used by any of the proposals. However, in this volatile multi-connectivity environment that involves the resolution of a complex optimization problem, the evaluation of the performance of such algorithms could be interesting, in particular for systems based on centralized control.

### 4.4. Multi-Connectivity to Improve Mobility Management

Efficient mobility management is a fourth objective associated with the use of multi-connectivity in 5G networks and Beyond (cf. Section 3.1.4). In this section, existing solutions are compared (cf. Section 4.4.1) and analyzed (Section 4.4.2).

#### 4.4.1. Existing Solutions to Improve Mobility Management

Different research projects have combined the ideas of mobility management and multi-connectivity: [9,97,112,113,114].

One of the first publications on the subject [9], demonstrated the benefits of an MC approach compared to traditional handover solutions. To achieve this, the authors proposed a comparison of the performance of MC and traffic-steering approaches by considering the relevant parameters for mobility, that is, Radio Link Failure (RLF), outage, and throughput. They also studied the impact of the MC system itself, taking into account different instantiation delays for the secondary base station, as well as the presence of small-cell 4G base stations co-located with the 5G base station.

In addition, various papers such as [112,113,114] focused on the use of the MC approach to improve mobility management. Among these different proposals, the authors of [112] proposed the simplest system, which used a deep-learning algorithm called Long Short-Term Memory (LSTM), to determine the mobility patterns of each UE and thus anticipate future movement trends. This information was then used to determine which base station should be used to establish a secondary connection with the UE. This approach provided interesting performance in terms of the throughput and handover prediction accuracy. However, the additional costs (calculation, latency) associated with the application to each UE of a deep learning-based solution and the impact of these additional costs were not evaluated. In [113], a richer model was proposed and three main contributions were introduced. First of all, instead of predicting the mobility of the UEs, the authors suggested measuring the channel quality at different base stations for each UE. As a result, depending on the actual network state, optimal associations were identified. They also recommended the use of a local coordinator to manage traffic between the different cells. Based on DC architecture, they finally defined a simplified handover procedure. This system provided attractive performance but its evaluation was only possible by considering a simulated semi-statistical channel model. Moreover, the integration of such a solution in the standardized DC architecture should be studied. In the same direction, the authors of [114] proposed a complete framework for MC primarily designed to limit the number of handovers. The first interesting idea introduced by these authors was to consider the different parameters to manage multi-connectivity, such as the received signal strength, UE velocity, 4G/5G channel states, 4G/5G signal path loss, and BS load. They also showed how their solution could be integrated into the reference architecture. The second important element in this paper was the definition of a handover management mechanism. This mechanism was based on a reward function derived from the parameters introduced above and a Markov decision process. This approach could be an effective way to manage mobility in 5G and future networks. Nevertheless, it might be interesting to compare this proposal with mechanisms based on intelligent approaches that could guarantee greater flexibility and reliability such as [112].

Finally, the authors of [97] looked at the application of multi-connectivity for future sliced 5G networks (cf. Section 2.2.5). More specifically, they proposed an improvement of the 3GPP EN-DC standard based on the LTE-LWA protocol, enabling the coordination of both 4G/5G bases stations and WLAN APs within the radio access network. This solution, completed with an evolved 3GPP UE session establishment call flow, was then used by these authors to define a protocol guaranteeing a higher availability and more efficient mobility management of UE slices. This solution, which extended the use of multi-connectivity to new areas (Network Slicing), had significant value. However, the proposed solution was simple and reactive, as slices were only deployed on user demand. As a result, the performance of such a solution would be limited and unsuitable for low-latency and high-reliability applications.

#### 4.4.2. Discussion Regarding Mobility Management

Research papers dealing with the use of multi-connectivity for mobility management have demonstrated the benefits of this approach [9] and introduced implementable algorithms [97,112] and frameworks [113,114]. These papers, which consider different scenarios (RATs, architecture) as well as different tools (AI, Markov, etc.), provide an interesting starting point for future research in this field (cf. Table 6). However, several limitations can be identified:Given the available tools (cf. Section 3.7), we can see that no framework has proposed a modular solution based on a high-performance predictive approach (cf. Table 6). This might appear as an interesting idea. Moreover, the level of performance (throughput, delay, packet loss) of such proactive MC systems should be evaluated.The number of RATs considered in mobility management has so far been limited. Indeed, apart from [97] that aimed to integrate WLAN APs, the other papers only focused on cellular RATs. The integration of satellite and WLAN RATs (outside the Network Slicing scope) or even LPWA RATs would therefore seem relevant for confirming the relevance of the MC approach.In the continuity of [97], it could also be interesting to propose other solutions using MC to manage UE slices. Indeed, this question is currently a key priority [115]. To achieve this and more broadly to manage mobility, it might be useful to consider the SDN approach. The technology, which guarantees a high level of flexibility, could be applied to the reduction of delays induced by mobility. Indeed, these delays lead to a considerable decrease in the benefits associated with the multi-connectivity approach [9].

### 4.5. Multi-Connectivity to Improve Spectrum and Interference Management

Efficient spectrum and interference management is the fifth objective associated with the use of multi-connectivity in 5G networks and beyond (cf. Section 3.1.5). In this section, the existing solutions are compared (cf. Section 4.5.1) and analyzed (Section 4.5.2).

#### 4.5.1. Existing Solutions to Improve Spectrum and Interference Management

Different research projects have combined the ideas of spectrum and interference management and multi-connectivity: [116,117,118,119,120,121,122].

For many of these studies, the objective was to determine how interference could be minimized to enable effective multi-connectivity in 5G cellular networks and beyond [116,117,119,120]. To achieve this, the authors of [116] introduced a user-centric approach called back-haul state-based distributed transmission. The proposed heuristic scheme in a dual-connectivity scenario, enables the UE to autonomously manage power transmission using two main criteria, the current back-haul links’ loads and the communication channels’ quality indicators. As demonstrated by the authors, this energy-efficient solution significantly reduced co-channel interference. However, this approach did not embrace the idea of cooperation between UEs although this could make it more effective. In addition, communication between one UE and two base stations was the only case considered. In the same way, the authors of [117] proposed an interesting nonlinear interference cancellation mechanism designed to enable multi-connectivity and higher throughput for IoT gateways. This solution was divided into two main phases. First, the harmonic interference and extra intermodulation components were represented using a Neural Network (NN) model. Then, this model was used by the receiver to extract these components and limit interference. An interesting aspect of this approach was that the authors proposed a hardware implementation demonstrating high performance. Nevertheless, the additional costs associated with the Neural Network (NN) model were not assessed. Moreover, as the authors focused on IoT gateways, the scope of application of such a solution is reduced. Finally, the authors of [119,120], involved in the European 5G-ALLSTAR project, dealt with cellular-satellite multi-connectivity. In this context, this paper presented three main contributions. First, frequency bands that could potentially be used to integrate satellite communications in 5G networks were identified. Then, a new channel modeling solution enabling the simulation of both satellite and cellular communications using the ray-tracing rendering technique was defined. Finally, three approaches were introduced for interference management using the application of a filtering technique at the transmitter side, an improved beam-forming technique, and a radio resource management algorithm. The approach proposed in these papers was interesting but the integration of such a solution in 5G networks will have to be studied. Moreover, it will also be necessary to ensure coexistence with other 5G RATs.

On the other hand, some studies focused on the potential benefits of multi-connectivity for spectrum and interference management [121,122]. The authors of [121] defined a new multi-connectivity architecture based on the selection of local anchors. This architecture was used to enable a soft and efficient handover. In this system, the selected anchors, which corresponded to femto-cell base stations, managed the surrounding cells called virtual cells and the UE–cell association. This multi-layered architecture improved UE mobility management and eliminate inter-cell interference. This user-centric approach showed interesting performance in terms of the EU spectrum efficiency and HOF rate. However, the additional costs associated with such a system, which involved a large number of exchanges between base stations, as well as the possibility of implementing such an architecture must be studied. The authors of [122] also investigated the use of virtualization and multi-connectivity to limit interference. In fact, considering that the Network Slicing architecture guaranteed the isolation between the different slices and therefore prevented inter-slices interference, the authors defined a solution based on this architecture. Then, taking into account users’ QoS requirements, they proposed a one-to-many matching game used to solve the user-to-slice association problem. This solution was promising for intra-cell interference management as inter-slice interference was avoided but it showed some significant limitations in terms of spectral efficiency and inter-cell interference management.

#### 4.5.2. Discussion Regarding Spectrum and Interference Management

Research papers have demonstrated how multi-connectivity operation could be improved thanks to efficient spectrum and interference management [116,117,119,120]. In contrast, other papers have proved that multi-connectivity could be useful for spectrum and interference management [121,122]. These papers, which considered different tools (AI, game theory) as well as different control approaches (UE, infrastructure) and different RATs (cellular, satellite), provide an interesting starting point for future research in this field (cf. Table 7). However, several limitations can be identified:Both infrastructure- and user-centric solutions have been considered to more efficiently manage spectrum and interference. However, hybrid approaches that combine EU-level and infrastructure-level decision making are increasingly being used nowadays, in particular, for inter-cell UL interference [123]. Therefore, it might be relevant to look at these approaches in a multi-connectivity scenario.As noticed by the authors of [122], Network Slicing architecture could be an efficient way to limit interference, as inter-slice isolation is ensured. However, in this paper, only intra-cell interference was studied. Therefore, it might be useful to examine the impact of Network Slicing on inter-cell interference management.Wireless back hauling, combined with an efficient beam-forming technique, is a promising way to minimize interference and increase spectrum efficiency in ultra-dense 5G networks [124]. Nevertheless, this idea has not been considered so far in a multi-connectivity scenario and interactions between base stations and multi-layered architecture could serve this idea.

## 5. Future Directions

As shown in Section 4, the application of multi-connectivity in 5G networks and beyond has already been considered for different purposes such as quality of service, energy efficiency, fairness, mobility management, and spectrum and interference management. However, some issues still need to be considered to provide a comprehensive solution and to take advantage of the enhancements enabled by 5G and future networks. Moreover, future directions can be imagined (cf. Figure 3).

### 5.1. Multi-Operator Architecture

Existing solutions assume that a single operator will support all the applications requested by end users (cf. Section 4). However, some applications, especially URLLC applications, may require a level of performance that a single operator may not be able to provide everywhere and at all times [125]. Furthermore, for these URLLC applications, road operators could deploy their WiFi networks to effectively manage their road infrastructure [126]. Therefore, for such scenarios (insufficient resources, diverse objectives), it would be necessary to consider a more realistic architecture, that is, a multi-operator architecture [127,128] (cf. Figure 3—Direction 1). For multi-connectivity, this idea of a multi-operator architecture is novel. Therefore, it is now necessary to design new solutions that will enable the efficient management of this multi-operator multi-connectivity. In particular, it could be interesting to consider approaches based on SDN, NFV, and C-RAN. Indeed, these centralized and software approaches have demonstrated their benefits for multi-connectivity in energy-efficiency [100] and fairness [107] contexts. Such architectures could provide transparent and efficient decision-making processes in multi-operator scenarios. It could also be interesting to explore improvements in these technologies (distribution, hierarchy, etc.) to ensure significant scalability.

### 5.2. Network Slicing for Future MC Services

Most solutions proposed in the literature such as [89,129] associate each user with a single set of QoS parameters such as bandwidth, latency, jitter, packet loss rate, etc. In other words, each user is associated with a single application and the user–cell association process aims to select accessible base stations meeting these requirements. However, a given user could simultaneously use applications of different types (enhanced mobile broadband, massive machine-type communications, URLLC) with different requirements. This would have an impact on the selection process. Indeed, a given base station/network node might meet the needs of one application but not those of another. For example, 4G base stations might not support URLLC applications but could offer a streaming service to the same user. Therefore, it seems important to leverage the Network Slicing benefits (cf. Section 2.2.5) and to apply this approach in a multi-connectivity scenario [97] (cf. Figure 3—Direction 2). Thus, the user–cell association should be performed not for each user but for each service (slice) required by that user. This could lead to more realistic and more efficient solutions optimally using multi-connectivity and meeting the 5G network objectives. To enable this, it could be relevant to consider the implementation of Artificial Intelligence-based solutions as these techniques (machine learning, deep learning) have demonstrated their benefits to solving optimization problems [84,112,117]. However, it will be essential in the future to study an element that has been ignored so far, that is, the additional costs (computation, latency) associated with the use of these approaches. To develop standardized solutions, it will also be necessary to rely on existing standards (LWA, CoMP, DC, etc.) and to develop solutions compatible with them.

### 5.3. Device-to-Device Relaying

The approaches proposed so far in multi-connectivity scenarios only consider direct communications between users and network nodes such as base stations, WiFi access points, etc. However, direct Device-to-Device (D2D) communications [130] are also an important feature of 5G and future networks. Indeed, these D2D communications could provide an interesting solution for offloading the core network. They could also be a way to extend network coverage through the definition of relay UEs that could be used by other out-of-coverage user equipment. Thus, D2D communications could improve both network coverage and load balancing between neighboring base stations as demonstrated in [131,132]. Therefore, in the multi-connectivity context, it might be interesting to consider these D2D communications (cf. Figure 3—Direction 3). Indeed, by using relay UEs, it may be possible to consider a larger number of network nodes for each UE and this could lead to improved user experiences and network utilization. However, the use of the communication resources of the relay UEs would involve increased power consumption for these UEs. Therefore, it could be useful to consider the selection of optimal relay nodes such as traffic lights, vehicles, etc. Indeed, for these UEs, unlike phones, energy resources are not a primary issue. Another potential direction could be to combine the ideas of D2D communications and Network Slicing and to identify for which applications (slices) these D2D communications could be used.

### 5.4. Service Continuity across Heterogeneous Wireless Networks

To meet the requirements of the URLLC and enhanced mobile broadband services, the deployment of edge nodes is a widely used solution today [133]. These edge nodes enable data processing to be carried out as close as possible to the users, guaranteeing low latency and high availability. However, different deployment solutions are considered for these edge nodes, both at the infrastructure level and within the UEs [134]. Moreover, in a multi-connectivity scenario, the same application could communicate simultaneously through different RATs potentially managed by multiple operators [135,136]. Edge-data processing in this environment is therefore a complex issue (cf. Figure 3—Direction 4). Indeed, to guarantee low latency, it might be necessary to communicate with the same edge server through different RATs or even with different edge servers managed by different operators. Therefore, it seems necessary to develop solutions enabling the global management of edge servers deployed at different levels (infrastructure, UE) and accessible through different RATs and different operators. To achieve this, various solutions appear to be relevant and have started to be explored in this context, notably, SDN [137,138] and AI techniques [139,140].

## 6. Conclusions

5G networks and beyond will have to cope with a growing number of connected devices, an ever-increasing bandwidth demand, and, at the same time, will have to support applications requiring low latency and high reliability. To achieve this, a new paradigm seems promising, that is, multi-connectivity. This approach enables the simultaneous connection of user equipment to several network nodes (base stations, WiFi access points, etc.) and could guarantee both better mobility management, higher throughput, and enhanced reliability.

Therefore, in this paper, we provide a comprehensive basis for future work on multi-connectivity in cellular networks. To achieve this, we first provide a comprehensive review of existing standards (cellular/cellular, cellular/WLAN) and enabling technologies (SDN, C-RAN, Network Slicing, etc.) for multi-connectivity. Then, we define a taxonomy aiming to classify the different elements characterizing multi-connectivity such as strategies, objectives, RATs, etc. Thereafter, we propose a comparison of existing MC-based applications for QoS improvement, energy efficiency, fairness, mobility, and efficiency management. We also highlight some lessons learned from the analysis of existing works such as architecture, technology, etc. Finally, we present some future directions for multi-connectivity in 5G networks such as a definition of multi-operator architecture, network slicing management, device-to-device communications benefits, and service continuity across heterogeneous wireless networks.

## Figures and Tables

**Figure 1 sensors-22-07591-f001:**
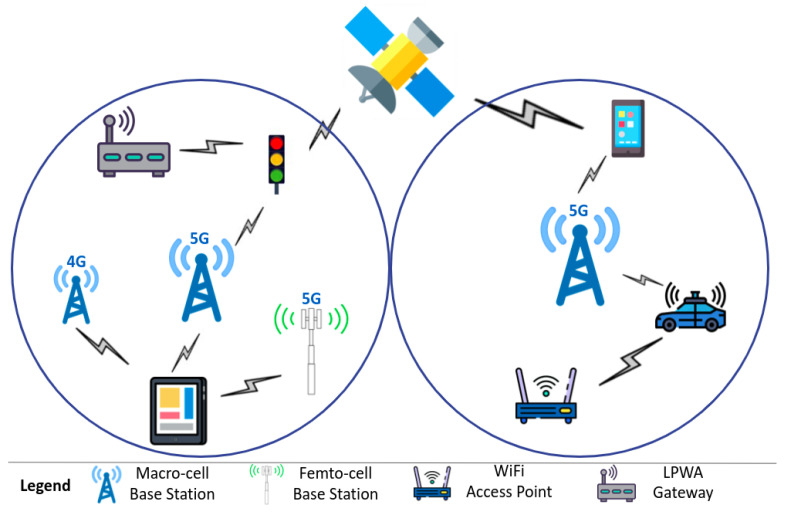
Presentation of different multi-connectivity scenarios for 5G Networks and beyond.

**Figure 2 sensors-22-07591-f002:**
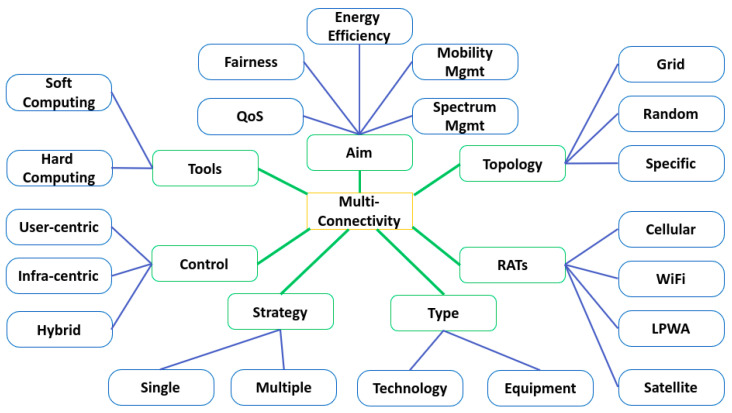
Taxonomy for multi-connectivity in 5G networks and beyond.

**Figure 3 sensors-22-07591-f003:**
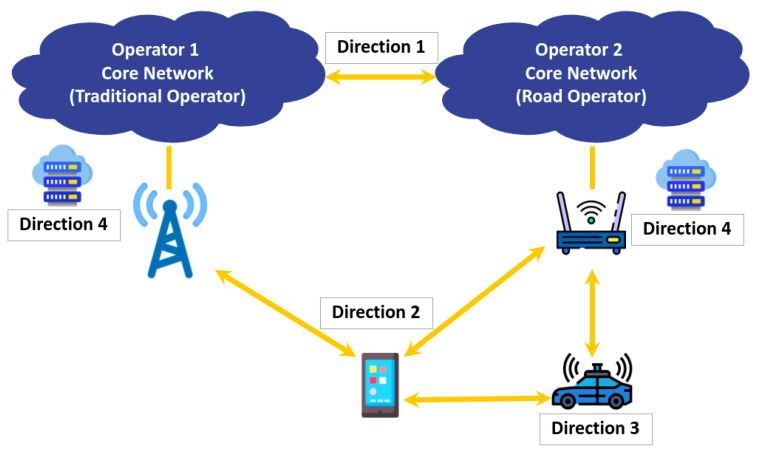
Presentation of potential future directions for multi-connectivity in 5G networks and beyond.

**Table 1 sensors-22-07591-t001:** Overview of existing standards for multi-connectivity in 5G networks and beyond.

Standard (Type)	Approach	Benefits (+) Limits (−)
CoMP (LTE-5G)	Direct coordination Utilization of a single carrier	**+** Interference reduction − Control overhead − Performance
DC (LTE-5G)	Presence of a master node Utilization of different carriers	**+** Performance **+** Control efficiency − Interference coordination
LTE-LWA (LTE-WLAN)	LTE-DC-like approach Per-packet routing	**+** High performance **+** Flexibility − WT required
LTE-LWIP (LTE-WLAN)	IPsec routed via WLAN Access Per-packet switching	**+** Simple deployment − Flexibility − Low performance

**Table 2 sensors-22-07591-t002:** Overview of enabling technologies for multi-connectivity in 5G networks and beyond.

Technology	Description	Benefits
SDN	Decoupling the network control and forwarding planes	Interoperability Load balancing Mobility management Overhead reduction
NFV	Decoupling network functions from proprietary hardware appliances	Cost reduction Mobility management Scalability
C-RAN	Decoupling BBUs from radio access units (BBUs pool)	Cost reduction Interoperability Scalability
CR	Ability to automatically adapt radio parameters according to the environment	Flexilibity Spectrum management User-centricity
NS	Ability to deploy independent virtualized networks over a single physical network	Flexilibity NFV benefits SDN benefits
AI	Decision support tools Ability to learn from experience	Load balancing Mobility management Spectrum management

**Table 3 sensors-22-07591-t003:** Comparison of existing solutions to improve QoS.

Proposition	Contributions	Limits	RATs	Tools
[89] Throughput improvement	MC problem formulation (NP-hard) Maximization of DL traffic	Unrealistic scenarios Inter-macro BSs handover	4G/5G	KKT
[90] Throughput improvement	Archi. for MmWave-MC Efficient BS selection for connectivity	Non-standalone Architecture Inflexible algo.	4G/5G	-
[91] Throughput improvement	Overcome MmWave limitations Efficient link scheduling	Control overhead Management overhead	4G/5G	-
[92] Latency reduction	Multi-RAT MC archi. SDN-based MC management	Absence of implementation	4G/5G WLAN	-
[93] Latency reduction	Trade-off between latency and resource MC (de)activation	Network load evaluation Single RAT	5G	Heuristic

**Table 4 sensors-22-07591-t004:** Comparison of existing solutions to improve energy efficiency.

Proposition	Contributions	Limits	RATs	Tools
[99] Reduce MC impact on global energy consump.	Efficient cell management (activation) Low-latency UE–cell association	Unrealistic scenarios Absence of mobility	4G/5G	Lyapunov opti.
[11] Reduce MC impact on global energy consump.	Combination of energy and network areas Smart Energy Management module	Simplistic solution Unrealistic architecture	4G/5G	-
[100] MC for energy efficiency	DL/UL traffic decoupling UE association problem formulation (NP)	Handover management Per-application slicing	4G/5G	LPR-R GAP
[12] MC for energy efficiency	Definition of MC scenarios Comparison of different algorithms	Optimization for single connectivity UE level evaluation	4G/5G	-
[104] Trade-off between perf. and energy efficiency	Multi-objective problem formulation Efficient load balancing	Unrealistic scenario Inter-BSs mobility	4G/5G	-
[101,102] Trade-off between perf. and energy efficiency	Efficient mobility management Accurate load level estimation	Insufficient evaluation of the impact of the proposed solution	4G/5G	-
[103] Trade-off between perf. and energy efficiency	Combination of energy and network areas Consideration of perf. degradation	Dynamic management Energy sharing	4G/5G	KKT

**Table 5 sensors-22-07591-t005:** Comparison of existing solutions for improving fairness.

Proposition	Contributions	Limits	RATs	Tools	Control
[107] MC for Proportional Fair	Efficient PF archi. definition Definition of heuristic algos for PF	Non-standardized architecture Overhead evaluation	4G/5G	Stable Matching Game	Centralized
[108] MC for Proportional Fair	Consideration of different scenarios Offloading of macro-cell BS	Non-standardized architecture Reactive solution	4G/WLAN	Water-Filling	Distributed
[109] MC for Proportional Fair	Definition of Minimum Rate Requirements Two stage iterative algorithm	Unrealistic scenario Scalar channel model	4G	Lagrange dual decomposition	Distributed
[110] MC for Proportional Fair	Local optimum calculation Joint cell select. and power control	Complex integration in standardized architecture Limited RATs	4G	Matching Game	Distributed

**Table 6 sensors-22-07591-t006:** Comparison of existing solutions for improving mobility management.

Proposition	Contributions	Limits	RATs	Tools
[9] Analysis of MC benefits	Demonstration of MC benefits Study of optimal params for MC	Unrealistic scenarios Limited to cellular RATs	4G/5G	-
[112] MC for mobility management (algorithm)	Mobility patters determination Efficient UE association	Overhead associated to the solution (computation) Overhead impact (latency)	4G/5G	Deep Learning
[113] MC for mobility management (framework)	Channel quality measurement Local coordinator definition for MC MC-based handover procedure	Unrealistic scenarios for simulation Complex integration in standardized architecture	4G/5G	Multiple-criteria decision-making
[114] MC for mobility management (framework)	Multi-criteria MC management Evaluation of the integration in the reference archi.	Lack of flexibility	4G/5G	Markov
[97] MC for Sliced 5G Networks (framework)	Extends DC to WLAN APs at RAN Extends MC to Network Slicing	Lack of flexibility Reactive solution (latency)	4G/5G/WLAN	-

**Table 7 sensors-22-07591-t007:** Comparison of existing solutions for improving spectrum and interference management.

Proposition	Contributions	Limits	RATs	Tools	Control
[116] Spect. management for MC	Multi-criteria optimization Energy-efficient solution	Non-cooperative approach Limited to two simultaneous connections	4G	-	UE
[117] Spect. management for MC	Two steps non-linear cancellation interference Hardware prototype implementation	Limited scenario Potentiel overhead	4G	Neural Networks	Infrastructure
[119,120] Spect. management for MC	Identification of frequency bands for satellite communications Channel modelling Multi solution for interference management	Integration in 5G architecture Coexistence with other RATs	5G Satellite	-	Infrastructure
[121] MC for spect. management	New multi-connectivity architecture Virtual cells definition	Archi. overhead Real-world deployment	4G/5G	-	UE
[122] MC for spect. management	Network slicing-based architecture Users to slice association	Spect. efficiency Inter-cell interference	5G	Matching game	Infrastructure

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
