# Peer review of "Multi-Connectivity for 5G Networks and Beyond: A Survey"

_sensors, 2022, doi:10.3390/s22197591_

Round 1

Reviewer 1 Report

This review article addresses the Multi-Connectivity for 5G Networks and beyond: a Survey. A comprehensive basis for future work on multi- connectivity in cellular networks. This is an interesting study and the authors have collected a unique dataset for 5G Networks and beyond. It is a well-written, needed, and useful summary of the current status of “data publication” from a particular perspective. Here are some minor comments:

#1 The authors need to be bolder and more analytical.

 #2: The author could discuss the specific complications/problem of the existing methods which should be given as a separate section.

#3: The author should remove or edit reference number 4. Research gate citations seem inappropriate.

Author Response

Dear Sir/Mme,

We thanks you for your interest towards our work. We have taken into account your revision requests in the final manuscript version.

As you had asked us, we improved the manuscript on the form (excessive repetition of certain terms, more consistent capitalization and naming, more relevant references) but also on the content of the document by adding some elements that seemed to be missing (explanation of the interest of AI, operation of NS in a multi-technology environment, jitter as QoS parameter, etc.).

We hope that the quality of this paper is now sufficient for publication in this journal.

Reviewer 2 Report

Summary

The paper offers an overview of many connection options for multi-connectivity. Less attention is paid to the lower layers, such as routing algorithms, but rather a more general overview of what is possible is given.
Multi-connectivity seems promising to cope with the growing numbers of devices and increasing demands for bandwidth and lower latencies in 5G and future networks.
Besides a summary of existing standards such as cellular networks and enabling technologies such as SDN and CRAN, a categorization of elements characterizing MC is done. For example energy consumption and fairness are discussed.

General Comments

- Great general overview of many possibilities for DC/MC including many access network technologies such as, for example, satellite-based communications
- Tables offer comparable data between many papers
- Table contents are not vertically aligned consistently (most times second or third column misaligned)
- in general many statements would benefit from one or two sentences more detailled explaination, e.g. explaination of AI improved cellular networks (page 8, line 291/292)
- by the nature of a survey, many technologies are introduced and not further explained, the reader could understand more if more details would be available. For example in line 281 (see detailed comments). Additionally some figures showing e.g. NS, Handovers (Xn) etc could help to understand the mentioned technologies/procedures better
- QoS is mentioned all times without jitter, the QoS metrics should include jitter

Detailed Comments

- “5G and beyond” occurs very often in text, maybe consider writing 5G and future networks or something similar
- “etc” occurs very often in the text – “among others” could be a substitute
- Line 275: NS is not a combination of SDN and VNF
- Line 281: It is unclear how 5G NS are enabling MC as it is only 5G, how NS@WiFi?
- Line 107 and 372: double point on left side
- Capitalization of abbreviations is not consistent (e.g. WiFi/Wifi, Master/master)
- Names not used consistently (LTE/4G, WiFI/WLAN)
- Chapter 3.1.2: It is mentioned that a 5G Handover procedure requires a lot of energy, it is not clear why the handover requires more energy compare to MC as for MC several connections to several cell-towers are run in parallel which for non-experts looks like it consumes more energy than just using one connection which can change (handover). 

Author Response

(The authors gave the same response as above.)
